# *Chlamydia* interfere with an interaction between the mannose-6-phosphate receptor and sorting nexins to counteract host restriction

Cherilyn A Elwell[1†], Nadine Czudnochowski[1†], John von Dollen[2],
Jeffrey R Johnson[2], Rachel Nakagawa[1], Kathleen Mirrashidi[1], Nevan J Krogan[2,3,4],
Joanne N Engel[1,5*], Oren S Rosenberg[1*]

[1]Department of Medicine, University of California, San Francisco, San Francisco,
United States; [2]Department of Cellular and Molecular Pharmacology, University of
California, San Francisco, San Francisco, United States; [3]QB3, California Institute for
Quantitative Biosciences, San Francisco, United States; [4]Gladstone Institutes, San
Francisco, United States; [5]Department of Microbiology and Immunology, University
of California, San Francisco, San Francisco, United States

*For correspondence: joanne.
engel@ucsf.edu (JNE); oren.
rosenberg@ucsf.edu (OSR)

†These authors contributed
equally to this work

Competing interests: The
authors declare that no
competing interests exist.

Reviewing editor: Suzanne R
Pfeffer, Stanford University
School of Medicine, United
States

**Abstract** *Chlamydia trachomatis* is an obligate intracellular pathogen that resides in a
membrane-bound compartment, the inclusion. The bacteria secrete a unique class of proteins, Incs,
which insert into the inclusion membrane and modulate the host-bacterium interface. We previously
reported that IncE binds specifically to the Sorting Nexin 5 Phox domain (SNX5-PX) and disrupts
retromer trafficking. Here, we present the crystal structure of the SNX5-PX:IncE complex, showing
IncE bound to a unique and highly conserved hydrophobic groove on SNX5. Mutagenesis of the
SNX5-PX:IncE binding surface disrupts a previously unsuspected interaction between SNX5 and the
cation-independent mannose-6-phosphate receptor (CI-MPR). Addition of IncE peptide inhibits the
interaction of CI-MPR with SNX5. Finally, *C. trachomatis* infection interferes with the SNX5:CI-MPR
interaction, suggesting that IncE and CI-MPR are dependent on the same binding surface on SNX5.
Our results provide new insights into retromer assembly and underscore the power of using
pathogens to discover disease-related cell biology.

## Introduction

*Chlamydia* species are obligate intracellular pathogens that are important causes of human disease
(*Mandell et al., 2010*). *C. trachomatis* infection is the most common bacterial sexually transmitted
disease and a leading cause of preventable infertility in the western world. Infections by *C. tracho-
matis* also lead to blinding trachoma, the major cause of non-congenital blindness in developing
nations. Understanding the pathogenesis of *C. trachomatis* infections is critical to the development
of a successful vaccine, which remains elusive (*Rey-Ladino et al., 2014*).

All *Chlamydia* species share a common intracellular developmental cycle in which they alternate
between an infectious, spore-like elementary body (EB), and a non-infectious, metabolically active
reticulate body (RB) (*Bastidas et al., 2013*; *Elwell et al., 2016*). Upon entry into non-phagocytic
cells, the EB resides within a membrane-bound compartment—the inclusion—and quickly diverges
from the canonical endo-lysosomal pathway. The EB differentiates into an RB and bacterial cell divi-
sion commences. After replicating within the ever-enlarging inclusion over a 24–72 hr time period,

the RB differentiates back to an EB. Mature EBs are released by host cell lysis or through an active extrusion process, ready to infect neighboring cells.

As obligate intracellular pathogens, *Chlamydiae* are absolutely reliant on altering host cell trafficking to scavenge nutrients while avoiding lysosomal fusion and recognition by the host innate immune system (*Bastidas et al., 2013*; *Elwell et al., 2016*). The *Chlamydia* inclusion maintains an intimate interaction with multiple host cell compartments and organelles, including the Golgi, endoplasmic reticulum (ER), mitochondria, endosomes, and lipid droplets (*Bastidas et al., 2013*; *Elwell et al., 2016*). However, the *Chlamydia* factors and host cell proteins that mediate these interactions are largely unknown. Likely bacterial candidates include the inclusion membrane proteins (Incs), a group of proteins that are unique to *Chlamydiales* and that are translocated from the bacteria through the Type III secretion system (T3SS) and inserted into the inclusion membrane (*Moore and Ouellette, 2014*). These proteins encode one or more bilobed hydrophobic domains composed of two closely spaced membrane-spanning domains, ~30 amino acids each, typically separated by a short loop region (*Bannantine et al., 2000*). Once inserted into the inclusion membrane, the Inc N- and C-terminal domains extend into the host cytoplasm (*Rockey et al., 2002*) and are thus ideally poised to modulate interactions at the host-pathogen interface.

Until recently, the host targets of only a few Incs had been identified (*Elwell et al., 2016*; *Moore and Ouellette, 2014*). We used high-throughput affinity purification-mass spectrometry (AP-MS) to globally define the Inc-human protein interactome and identified putative binding partners for 38/58 of the *C. trachomatis* Incs (*Mirrashidi et al., 2015*). One of the most robust interactions was between IncE, an early expressed Inc (*Scidmore-Carlson et al., 1999*), and a subset of retromer components. The retromer is a phylogenetically conserved multi-subunit complex that mediates retrograde transport of transmembrane protein cargo, such as the well-studied cation-independent mannose-6-phosphate receptor (CI-MPR), from endosomes to the *trans*-Golgi network (TGN) (reviewed in [*Liu, 2016*]). Retromer is composed of two distinct sub-complexes of tightly assembled subunits: a cargo recognition trimer (VPS26/29/35) and a membrane deforming heterodimer made up of sorting nexin (SNX) SNX1 or SNX2 paired with either SNX5 or SNX6. In mammals, as opposed to yeast, the association between the VPS trimer and the SNX sub-complexes is transient. The individual SNX1/2/5/6 proteins encode a Phox-homology (PX) domain that recognizes and binds to specific phosphoinositides on membranes and a Bin-amphiphysin-Rvs (BAR) domain that both senses and induces membrane curvature. These SNX-BAR proteins facilitate formation of tubules into which the cargo receptors are sorted. We recently discovered that *C. trachomatis* IncE binds directly to the PX domains of SNX5/6 but not SNX1/2 and that IncE is sufficient to disrupt retromer-dependent trafficking of CI-MPR (*Mirrashidi et al., 2015*). In addition, we and others found that SNX-BAR proteins relocalize from endosomes to the inclusion membrane during *C. trachomatis* infection and that depletion of SNX5/6 enhances *C. trachomatis* replication (*Aeberhard et al., 2015*; *Mirrashidi et al., 2015*). Our findings suggest that IncE binding to SNX5/6 redirects SNX1/2/5/6 away from their normal functional location and thereby limits the ability of retromer to control *C. trachomatis* infection (*Mirrashidi et al., 2015*). However, the molecular details by which IncE binds to the SNX5/6-PX domain are unknown; for example, does IncE binding form a novel interface, and does IncE displace a normal host binding partner? The mechanistic implications of this interaction are critical to our understanding of retromer function in pathogenesis.

In this work, we solved a high-resolution co-crystal structure of the C-terminus of IncE bound to the PX domain of SNX5. We demonstrate that IncE binds to a previously unappreciated highly conserved hydrophobic groove in the SNX5-PX domain and that the binding site is distinct from the SNX5 phosphoinositide-binding site. Using quantitative AP-MS of wild type (WT) SNX5 and SNX5 mutated in key conserved residues in the hydrophobic groove, we identified an unsuspected interaction between SNX5-PX and CI-MPR. Together with direct biochemical assays and in vivo infection assays, we demonstrate that IncE disrupts the SNX5-PX:CI-MPR interaction to alter retromer-dependent trafficking. Modulation of retromer trafficking by this intracellular pathogen may facilitate its ability to survive within the hostile intracellular environment.

# Results

## The structure of the SNX5-PX$_{20-180}$:IncE$_{108-132}$ complex

To uncover the molecular basis for the strong interaction between IncE and SNX5 and to illuminate the biological function of this interaction, we determined the crystal structure of the *Mus musculus* SNX5-PX domain (SNX5-PX$_{20-180}$) bound to the C-terminal 25 amino acids of IncE (IncE$_{108-132}$) (*Figure 1*). The crystal structure was solved by molecular replacement at a resolution of 2.31 Å, using the SNX5-PX apo-structure as a model (PDB ID 3HPC) ([*Koharudin et al., 2009*] and *Supplementary file 3*). The structure contained four copies of the SNX5-PX:IncE complex in the asymmetric unit, which are all less than 0.41 Å RMSD between the molecules and do not appear different in conformation (*Figure 1—figure supplement 1A*). The discussion below refers to SNX5-PX chain A and its associated IncE (chain P) of the structure but could apply to any of the other chains. Inspection of $F_o$-$F_c$ maps indicated strong electron density in the structure that was absent in the 3HPC model (*Figure 1—figure supplement 2*). Using bulky hydrophobic residues as a guide, we were able to unambiguously place the IncE$_{108-132}$ peptide into the density, although amino acids 108–110 and 132 were disordered in the crystal and could not be built. The final model of the complex ($R_{work}$/$R_{free}$ 20.32/26.00) showed that IncE$_{108-132}$ forms a β-hairpin that does not appear to be externally stabilized by its interaction with the PX domain (*Figure 1A*). When compared to the apo-SNX5-PX structure (PDB ID 3HPC [*Koharudin et al., 2009*]), IncE$_{108-132}$ was found to bind in a deep, hydrophobic groove that is approximately 10 Å x 18 Å in area running perpendicular to the long axis

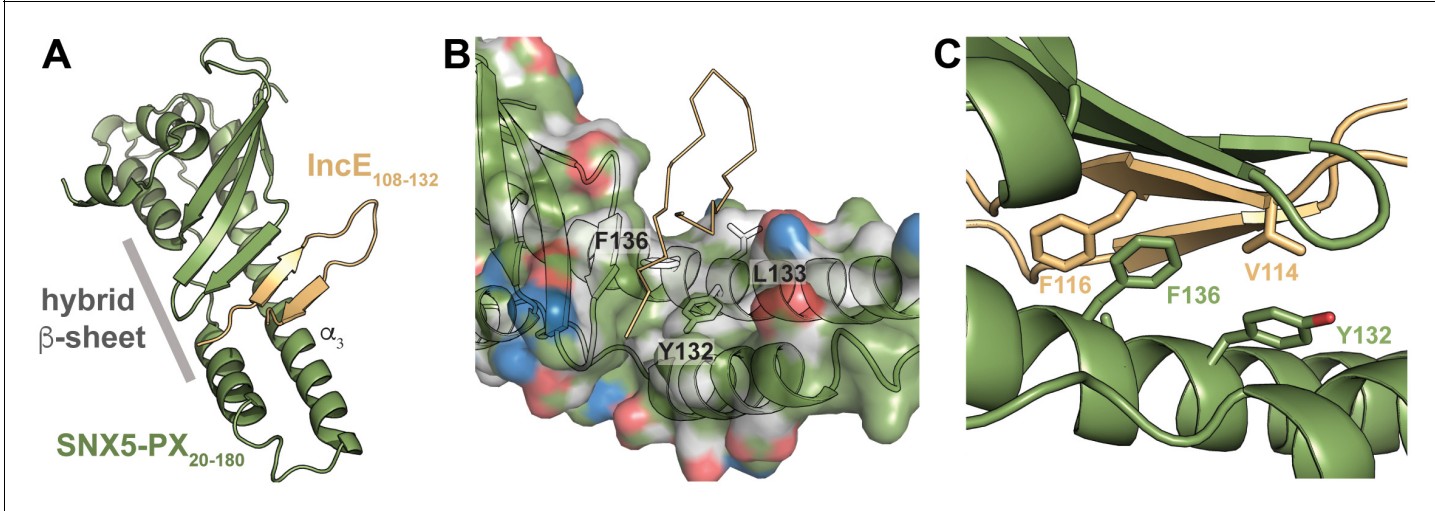

**Figure 1.** IncE binds in a hydrophobic groove on SNX5-PX, extending the SNX5-PX β-sheet. (**A**) The structure of SNX5-PX$_{20-180}$ (green) bound to IncE$_{108-132}$ (gold). Binding of IncE to SNX5 leads to the formation of a hybrid β-sheet. (**B**) Surface and ribbon representation of SNX5-PX showing the hydrophobic binding groove. Atoms are colored according to the scheme described in (*Hagemans et al., 2015*) to highlight hydrophobic surfaces. Carbon atoms not bound to nitrogen or oxygen atoms are colored grey, oxygens carrying the negative charges in glutamate and aspartate are red and nitrogens carrying the positive charges in lysine and arginine are blue, while all remaining atoms are green. The IncE interacting residues are shown as sticks and labeled. IncE is displayed as gold ribbon. (**C**) Close-up view of the SNX5-PX$_{20-180}$:IncE$_{108-132}$ binding surface with interacting residues shown as sticks.

The following figure supplements are available for figure 1:

**Figure supplement 1.** Structural analysis of the SNX5-PX:IncE complex.

**Figure supplement 2.** Difference density map reveals IncE$_{108-132}$.

**Figure supplement 3.** The hydrophobic binding groove of SNX5-PX is distant from the PIP binding groove.

**Figure supplement 4.** Overlay of the SNX5-PX:IncE structure with the structure of the SNX9-PX-BAR domain.

of the SNX5-PX domain. SNX5-PX$_{20-180}$ is a mixed α-β structure with three antiparallel β-sheets running along the surface of a long α-helix (α3) stabilized by two shorter helices. Two of the helices extend past the end of the β-sheets to form a long α-helical hairpin not seen in the crystal structures of other SNX-PX domains (*Koharudin et al., 2009*). In the SNX5-PX$_{20-180}$:IncE$_{108-132}$ structure, the SNX5-PX$_{20-180}$ β-sheet is completed by the IncE$_{108-132}$ β-hairpin, which binds on the surface of the α-helical hairpin in the hydrophobic groove, hydrogen bonding with the SNX5-PX$_{20-180}$ β-sheet to form a hybrid, five-stranded antiparallel β-sheet (*Figure 1A*). Outside of this β-sheet formation, the structure of SNX5 complexed with IncE$_{108-132}$ is notable for its almost complete lack of hydrogen bonding or ionic charge-charge interactions (*Figure 1B*), relying instead on the formation of a small hydrophobic core at the interface of the two proteins. This core buries five residues at the interface: Y132, L133 and F136 from SNX5-PX$_{20-180}$ and V114 and F116 from IncE$_{108-132}$ (*Figure 1C*).

## The IncE binding groove is highly conserved, suggesting a native protein-protein interaction surface

PX domain proteins bind to phosphatidylinositol phosphate (PIP) head groups (*Teasdale and Collins, 2012*). For example, the SNX5 PIP binding site binds specifically to phosphatidylinositol 4,5-bisphosphate (PI(4,5)P$_2$), although the binding is weak (*Koharudin et al., 2009*). In the SNX5-PX$_{20-180}$:IncE$_{108-132}$ crystal structure, we observed no direct interaction between IncE$_{108-132}$ and the previously observed PIP binding pocket (*Figure 1—figure supplement 3A*). In order to determine if binding of IncE altered the conformation of the SNX5-PX domain, we overlaid the apo- (PDB ID 3HPC) and IncE-bound structures and mapped RMSD values onto the surface for comparison (*Figure 1—figure supplement 1B*). The changes between the apo-SNX5-PX structure and the IncE$_{108-132}$ bound SNX5-PX$_{20-180}$ structure are small and consist mainly of an inward rotation of the SNX5 α-helical hairpin ~30° toward the IncE peptide. This region appears to be quite flexible as it has the highest B-factors in both the apo- and IncE-bound structures (*Koharudin et al., 2009*). We directly tested whether IncE$_{108-132}$ allosterically interfered with binding of PIPs by assaying the ability of SNX5-PX$_{20-180}$ to bind to immobilized phospholipids in the presence and absence of IncE$_{108-132}$. IncE$_{108-132}$ did not compete with binding of SNX5 to lipid head groups nor did it affect the pattern of PIPs engaged by SNX5-PX$_{20-180}$ (*Figure 1—figure supplement 3B*). Together, this data supports the notion that binding of IncE to SNX5-PX does not interfere with SNX5 binding to PIPs.

No structure is available of the full-length SNX5 (SNX5$_{FL}$) protein with the BAR domain intact; however, there are several other structures of SNX-BAR domain proteins (*Pylypenko et al., 2007*; *Wang et al., 2008*) that allow us to model the position of the BAR domain relative to SNX5-PX$_{20-180}$ (*Figure 1—figure supplement 4*). IncE$_{108-132}$ does not bind at this BAR domain interface, and the BAR domain interface does not appear to be significantly altered by binding of IncE. Although SNX5-PX$_{20-180}$ exhibits a unique structure that differs from other PX proteins, the regions that distinguish the SNX5-PX$_{20-180}$ structure are distant from the BAR domain interaction site. We conclude that the interaction between SNX5 and SNX1 through their respective BAR domains would not be affected by interaction of SNX5 with IncE.

A phylogenetic analysis of 763 unique sequences of SNX5, SNX6, and SNX32 allowed comparison of the IncE binding surface of each species and revealed that the three residues buried at the IncE$_{108-132}$ interface are identical in all species of metazoans for which sequence data is available (*Figure 2A*). This conservation was observed even in the most distantly related sequence we analyzed, SNX6 from *Trichinella spiralis*, which shows overall identity of only 49%. This region of conservation in the SNX5/6/32 family extends outside of the hydrophobic groove that binds to IncE and includes several solvent exposed residues, including K138 (*Figure 2B*). Importantly, these residues are absent in other SNX-BAR proteins including SNX1 and SNX2, as has been previously noted (*Koharudin et al., 2009*). Thus, the residues lining the SNX5 hydrophobic groove are not involved in the core folding of the SNX5/6/32-PX domain. These residues are solvent exposed in the unbound crystal structure (*Koharudin et al., 2009*), and yet they are identical throughout metazoan evolution, implying an essential function. Together, our finding that the IncE binding surface is highly conserved in the SNX5/6/32 family of proteins suggests that it serves a normal physiological role, most likely by mediating an interaction with another macromolecule.

**Figure 2.** The IncE binding residues are highly conserved among the SNX5/6/32 family of proteins and implicated in protein-protein interactions. (**A**) Alignment of selected SNX family members showing the conservation of the IncE-interacting residues Y132, L133 and F136 (red triangles) in SNX5/6/32 family members and their divergence in related SNX-BAR proteins. The only invariant residues in all SNX5/6/32 sequences are highlighted in pink. (**B**) Structural representation of the invariant residues (pink) shown in A. (**C**) HEK293T cells were transiently co-transfected with the indicated full-length FLAG-tagged $SNX_{WT}$ constructs and with Strep-tagged $IncE_{101-132}$. Lysates were immunoprecipitated with anti-FLAG beads and immunoblotted with the indicated antibodies. Input represents 1% of lysates. SNX15 serves as a negative control. The data shown is representative of two independent biological experiments.

The following figure supplement is available for figure 2:

**Figure supplement 1.** The hydrophobic binding groove of SNX5-PX is required for recruitment of SNX5 to *C. trachomatis* inclusions.

Biophysics and Structural Biology | Microbiology and Infectious Disease

## Mutation of the SNX5-PX:IncE binding surface disrupts IncE binding and SNX5 recruitment to the *C. trachomatis* inclusion

The crystal structure predicts that disruption of the small hydrophobic core at the SNX5-PX$_{20-180}$: IncE$_{108-132}$ interface would destabilize the complex. In particular, SNX5$_{Y132}$:IncE$_{V114}$ and SNX5$_{F136}$: IncE$_{F116}$ form stable van der Waals and π-π stacking interactions that appear to be central to the specificity of the interaction. We mutated SNX5 residues Y132 and F136 to aspartic acid or to asparagine and co-transfected the WT or mutated full-length FLAG-SNX5 constructs together with IncE$_{101-132}$-Strep into HEK293T cells. The SNX5 variants expressed at similar levels to SNX5$_{WT}$ (*Figure 2C*). Co-immunoprecipitation of IncE$_{101-132}$ with SNX5 was dependent upon Y132 or F136, confirming the importance of these aromatic amino acid contacts in the formation of the SNX5:IncE complex (*Figure 2C*). We previously demonstrated a robust recruitment of full-length FLAG-SNX5$_{WT}$ to the *C. trachomatis* inclusion and associated tubules (*Mirrashidi et al., 2015*). We next determined whether the IncE binding groove is necessary for recruitment of SNX5 to the *C. trachomatis* inclusion by analyzing the localization of FLAG-SNX5$_{Y132D,F136D}$ during infection. FLAG-SNX5$_{Y132D,F136D}$ fails to localize to the inclusion (*Figure 2—figure supplement 1*), although endogenous SNX6 was still recruited. Together, these data provide further support that IncE interacts with the SNX5 hydrophobic binding groove.

## Differential mass-spectrometry reveals a native interaction between SNX5-PX and CI-MPR

Given the high conservation of the IncE binding surface, we reasoned that IncE is likely disrupting an interaction between SNX5 and a previously unappreciated host cell protein through a shared binding surface. Thus, we hypothesized that mutating this surface would lead to loss of the native interaction. We performed AP-MS to identify host cell proteins whose binding to SNX5 is dependent on the presence of the same hydrophobic residues, Y132 and F136, as are required for IncE binding. Full-length FLAG-SNX5$_{WT}$, FLAG-SNX5$_{Y132D}$ or FLAG-SNX5$_{F136D}$ were transfected into HEK293T cells. Affinity-purified eluates were analyzed by label-free, shotgun liquid chromatography-mass spectrometry/mass spectrometry (LC-MS/MS), employing a pipeline similar to our previous studies (*Jäger et al., 2011a*; *Mirrashidi et al., 2015*) (*Supplementary file 1*). Using SAINT-Express (*Teo et al., 2014*), we compared the protein-protein interactions of affinity purified lysates from untransfected cells to those prepared from SNX5$_{WT}$, SNX5$_{Y132D}$, or SNX5$_{F136D}$ transfected cells. We considered only those proteins that had a Bayesian False Discovery rate (BFDR) ≤ 0.1, a conservative threshold for the elimination of false positive interactions. Although six proteins were found to significantly interact with SNX5$_{WT}$ protein according to the SAINT algorithm, only two proteins, CI-MPR and Insulin-like Growth Factor I (IGF1R), bound differentially to SNX5$_{WT}$ and the SNX5 hydrophobic groove mutants, as evidenced by significantly different SAINT scores (*Table 1*, *Supplementary file 2* and *Figure 3*). We focused on CI-MPR because it is the most well studied retromer cargo and because we have previously established that ectopic expression of IncE is sufficient to disrupt CI-MPR trafficking (*Mirrashidi et al., 2015*). Importantly, SNX1 co-affinity purified with both WT and mutant SNX5 (*Supplementary file 2* and *Figure 3*), consistent with the notion that disrupting the IncE binding surface does not cause major alterations to the SNX5 structure. This result also suggests that the association of CI-MPR with SNX5 observed in the AP-MS analysis is not solely dependent on the previously described interaction of SNX1/2 with the VPS complex (*Haft et al., 2000*; *Rojas et al., 2007*).

## IncE peptide interferes with CI-MPR binding to the SNX5-associated complex

We have previously shown that expression of the C-terminus of IncE results in loss of CI-MPR co-localization with TGN-46 positive compartments and increases localization with VPS35 positive compartments (*Mirrashidi et al., 2015*). We next assessed whether IncE$_{108-132}$ could displace CI-MPR binding to SNX5-containing complexes. Lysates from HEK293T cells transfected with full-length FLAG-SNX5$_{WT}$ were mixed with increasing amounts of IncE$_{108-132}$, added to anti-FLAG beads, and eluates were assessed for CI-MPR co-immunoprecipitation. We also tested a mutant version of IncE$_{108-132}$ in which V114 and F116 were changed to asparagine and aspartic acid (IncE$_{V114N,F116D}$), which would not be predicted to disrupt SNX5:IncE or SNX5:CI-MPR complexes. In control

**Table 1.** SNX5 protein-protein interactions and their dependence on the IncE binding groove. Shown is the average of the spectral counts from the listed proteins detected in the affinity purified lysates prepared (in triplicate) from untransfected cells (mock) or from cells transfected with full-length $SNX5_{WT}$, $SNX5_{F136D}$, or $SNX5_{Y132D}$ mutants. Only interactions exhibiting a BFDR $\leq$ 0.1 when compared to affinity purified lysates from untransfected cells are listed. The proteins that have a SAINT score (**Teo et al., 2014**) of 1 (the highest probability possible) are highlighted in green and the SAINT score is shown in parentheses. Of the proteins found to interact with $SNX5_{WT}$, CI-MPR and IGF1R show a difference in their SAINT score between the WT and the mutants, suggesting that the mutations block the protein-protein interaction. (ND, not determined due to lack of spectral counts in data set).

| ProtID | Name | Mock | SNX5 WT | SNX5 F136D | SNX5 Y132D |
|---|---|---|---|---|---|
| Q9Y5X3 | SNX5_HUMAN | 4.33 | 134 (1) | 118.33 (1) | 121.67 (1) |
| Q13596 | SNX1_HUMAN | 0 | 70 (1) | 64 (1) | 64 (1) |
| O60749 | SNX2_HUMAN | 1 | 105.5 (1) | 101 (1) | 99.67 (1) |
| P11717 | CI-MPR_HUMAN | 0 | 12.67 (1) | 0.33 (0) | 0 (0) |
| P08069 | IGF1R_HUMAN | 0 | 8.67 (1) | ND | ND |
| P30048 | PRDX3_HUMAN | 0.33 | 4.67 (0.65) | 0.33 (0) | 1.33 (0.07) |
| Q15773 | MLF2_HUMAN | 0 | 1.67 (0.33) | 5.33 (0.98) | 2.67 (0.56) |

experiments, we performed an in vitro competition assay to demonstrate that WT $IncE_{108-132}$ peptide disrupts binding of 6xHis-MBP-$IncE_{101-132}$-Strep to 8xHis-SNX5-$PX_{20-180}$ (**Figure 4—figure**

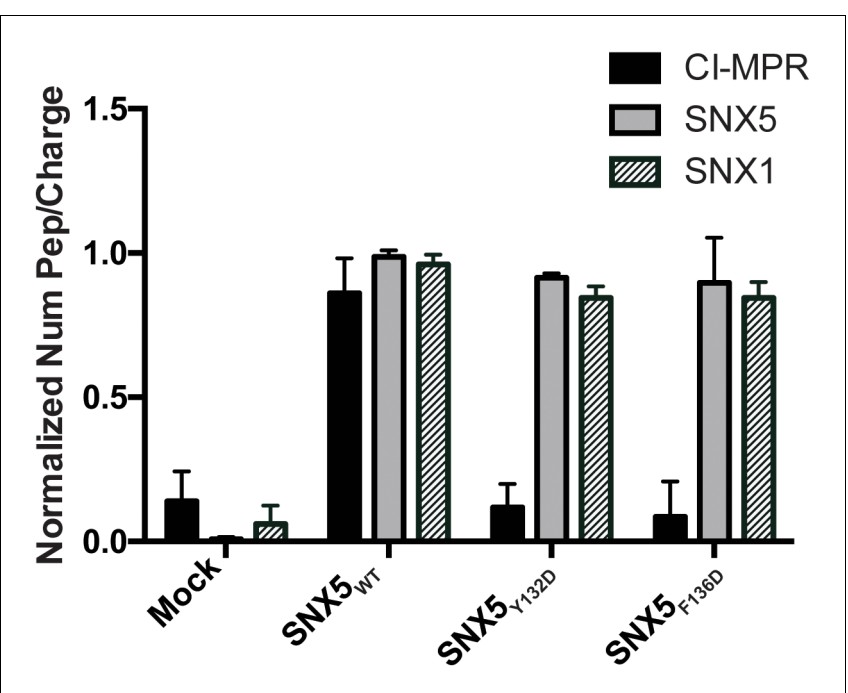

**Figure 3.** Comparative AP-MS for transfected $SNX5_{WT}$, $SNX5_{Y132D}$, and $SNX5_{F136D}$. The Y-axis quantifies the number of peptides/charge detected by LC-MS/MS for CI-MPR, SNX5, and SNX1 that affinity purified with the indicated full-length SNX5 proteins. Mock refers to untransfected cells. The graph is normalized to $SNX5_{WT}$. Whereas SNX1 is equally well represented in the affinity-purified lysates from WT and mutant SNX5, the representation of CI-MPR peptides differed significantly between $SNX5_{WT}$ and each of the SNX5 mutants. Data are mean +/− SD from three independent biological experiments.

*supplement 1*). Addition of WT IncE$_{108-132}$ to SNX5-containing complexes from HEK293T cells diminished SNX5:CI-MPR binding in a dose-dependent manner without affecting SNX1 binding (*Figure 4*). In contrast, the mutant IncE peptide failed to disrupt CI-MPR interactions with SNX5-containing complexes (*Figure 4*). These experiments show qualitatively that the C-terminus of IncE is capable of disrupting the SNX5:CI-MPR interaction in a manner that depends on IncE residues V114 and F116. We conclude that the IncE binding groove is required for the SNX5:CI-MPR interaction.

## Chlamydial infection disrupts CI-MPR association with SNX5 complexes

Based on our crystal structure, we predicted that *C. trachomatis* infection should therefore interfere with the CI-MPR:retromer interaction. We tested this notion by assessing CI-MPR co-immunoprecipitation with SNX5 during *C. trachomatis* infection (*Figure 5A*). As we have previously reported (*Mirrashidi et al., 2015*), IncE co-immunoprecipitates with FLAG-SNX5 in infected cells, and infection does not appreciably diminish steady-state levels of SNX1, VPS35, or CI-MPR. We did not observe a stable interaction between FLAG-SNX5 and VPS35 in either uninfected or infected cells, consistent with their known transient interaction (*Seaman, 2012*; *Swarbrick et al., 2011*). Remarkably, infection decreased the co-immunoprecipitation of CI-MPR with FLAG-SNX5 with minimal effects on the co-immunoprecipitation of SNX1 with FLAG-SNX5. We quantified the effect of infection on co-immunoprecipitation of CI-MPR with SNX5 by calculating the ratio of CI-MPR in infected versus uninfected FLAG-SNX5 transfected lysates and the ratio of CI-MPR in infected versus uninfected FLAG-SNX5 co-immunoprecipitations. We found an ~5.6 fold decrease in the ratio in co-immunoprecipitated complexes (*Figure 5B*). This result is consistent with a model in which IncE displaces CI-MPR from SNX5-containing complexes during infection.

## Discussion

Intracellular pathogens alter host cell trafficking pathways in order to survive in the hostile intracellular environment (*Kumar and Valdivia, 2009*). We have previously shown that the C-terminus of IncE binds directly to the PX domain of the retromer components SNX5 and SNX6, but fails to bind to the closely related SNX1-PX and SNX2-PX domains (*Mirrashidi et al., 2015*). By solving the crystal structure of the IncE-C-terminus in complex with the SNX5-PX domain, we demonstrate that IncE binds to a hydrophobic groove that is highly conserved among SNX5/6/32 homologs but absent in

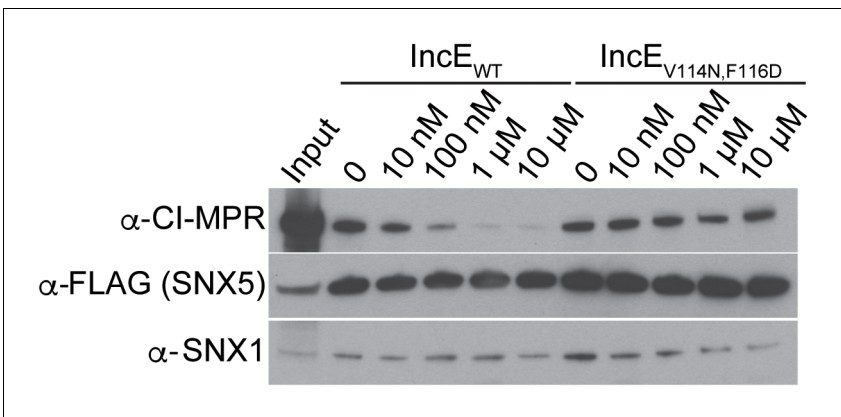

**Figure 4.** IncE$_{108-132}$ interferes with CI-MPR binding to SNX5-associated complexes. Lysates from HEK293T cells transiently expressing full-length FLAG-SNX5$_{WT}$ were incubated with anti-FLAG beads in the presence of the indicated concentrations of wild type or mutant IncE$_{108-132}$ under non-equilibrium conditions. Eluates were immunoblotted with the indicated antibodies. Input represents 1% of lysates. WT but not mutant IncE interferes with binding of CI-MPR to FLAG-SNX5-containing complexes. The data shown is representative of three biological experiments.

The following figure supplement is available for figure 4:

**Figure supplement 1.** IncE peptide interferes with IncE$_{101-132}$ binding to SNX5-PX$_{20-180}$ in vitro.

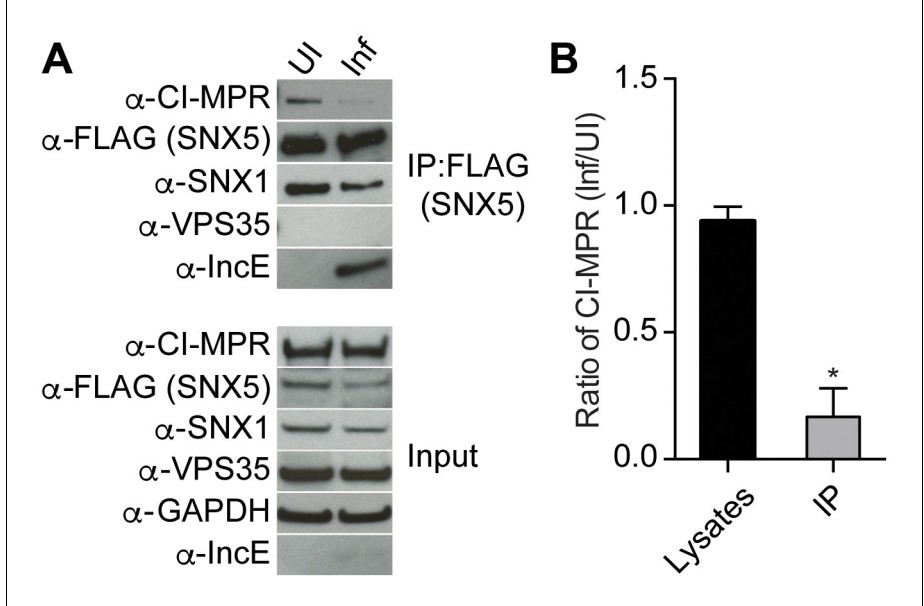

**Figure 5.** *C. trachomatis* infection interferes with CI-MPR binding to SNX5 complexes. (**A**) HeLa cells were transiently transfected with full-length FLAG-SNX5$_{WT}$ for 24 hr and then were infected (Inf) or left uninfected (UI) with *C. trachomatis* for 24 hr. FLAG-SNX5$_{WT}$ was immunoprecipitated and eluates were immunoblotted with the indicated antibodies. Input is 1% of lysates used for immunoprecipitation. (**B**) Quantitation of the ratio of CI-MPR from infected to uninfected cells in lysates or in eluates from FLAG-SNX5 immunoprecipitations (IP). Data are mean +/− SD from three independent biological experiments. *p=0.0004 compared to lysates, unpaired t-test.

SNX1/2 homologs, thus explaining the exquisite specificity of the IncE:SNX interaction. The high conservation of this SNX5/6/32 hydrophobic groove further suggests that IncE may be displacing or mimicking a host binding partner that is essential for host function. We therefore employed an unbiased quantitative AP-MS approach to identify host proteins whose interaction with SNX5-containing complexes was dependent upon an intact IncE binding surface. Unexpectedly, this analysis identified the CI-MPR, the most well-characterized transmembrane receptor sorted by the mammalian retromer. We confirmed in vivo that CI-MPR co-immunoprecipitates with ectopically-produced SNX5 and that co-immunoprecipitation is diminished in the IncE binding surface mutants. An IncE peptide corresponding to residues 108–132 inhibited binding of the CI-MPR to affinity-purified SNX5-containing complexes, whereas a mutant peptide altered in the predicted SNX5 binding residues was without effect. Finally, we demonstrate that *C. trachomatis* infection disrupts the association of CI-MPR with SNX5-associated complexes and that the SNX5 hydrophobic groove is required for its interaction with the inclusion. We conclude that the highly conserved IncE binding surface on SNX5 is required for its interaction with the CI-MPR. Based on our results, we favor the hypothesis that during *C. trachomatis* infection, inclusion-localized IncE displaces a lower affinity SNX5:CI-MPR interaction. From these studies, we can mechanistically link IncE to *Chlamydia*-mediated disruption of retromer trafficking, and thereby illuminate a new mechanism by which a vacuolar pathogen avoids retromer-mediated restriction of its intracellular developmental cycle.

Our most striking and unexpected result is that the conserved IncE binding surface on SNX5 was required for the association of the CI-MPR with retromer, as this membrane receptor is thought to interact with retromer through VPS35, which in turn interacts with SNX1/2 (*Arighi et al., 2004*; *Rojas et al., 2007*). Prior to solving the co-crystal structure, we predicted that IncE binding to SNX5 would affect the interaction of SNX5 with its known binding partners, SNX1/2 BAR domains or phosphoinositides (*Cullen and Carlton, 2012*; *Seaman, 2012*). However, our structure revealed no major conformational changes in the PX domain or the BAR domain-interacting surface. This finding is consistent with our observation that IncE peptide does not interfere with SNX5 binding to immobilized phospholipids or to SNX1. Our studies thus identify a previously unrecognized and functionally

important surface of SNX5 that may provide an additional level of cargo specificity during retrieval of CI-MPR (*Liu, 2016*).

The requirement of the hydrophobic groove in SNX5:CI-MPR interactions may be explained by several models. CI-MPR, and perhaps other retromer cargo, may bind directly to this site, which could serve as an additional low-affinity binding site for CI-MPR or CI-MPR may be sequentially transferred to SNX5 once it has been selected for retrieval by VPS. Alternatively, retromer cargo binding to SNX5 may be indirect, mediated by one or more additional protein(s). Finally, the hydrophobic binding groove could contribute to SNX5 association with membranes that contain the CI-MPR, leading to co-purification of the two proteins. Work is ongoing to differentiate these models.

Our work adds to our understanding of how intracellular pathogens modulate the retromer. Human Immunodeficiency Virus, Human Papilloma Virus, *Coxiella burnetii,* and *Salmonella typhimurium* require retromer function for efficient intracellular survival (*Bujny et al., 2008*; *Groppelli et al., 2014*; *Lipovsky et al., 2013*; *McDonough et al., 2013*; *Personnic et al., 2016*). In contrast, *Legionella pneumophila* infection inhibits retromer function through a secreted effector, RidL, which binds to VPS29 and PI(3)P (*Finsel et al., 2013*). Our results provide important mechanistic insights as to how *Chlamydia* interferes with host cell trafficking pathways and provide possible explanations for the intriguing result that depletion of SNX5/6 during *C. trachomatis* infection leads to increased infectious progeny (*Aeberhard et al., 2015*; *Mirrashidi et al., 2015*). CI-MPR plays a role inlysosomal function by virtue of its ability to transport newly synthesized acid hydrolase precursors from the TGN to endosomes for eventual delivery to lysosomes (*Bonifacino and Hurley, 2008*). RNAi-mediated depletion of retromer components prevents retrieval of acid hydrolase receptors to the TGN and results in their missorting to lysosomes and eventual degradation (*Arighi et al., 2004*). Mislocalization of CI-MPR also causes mistrafficking of Niemann-Pick Disease Type C2 Protein (NPC2), resulting in cholesterol accumulation in the late endosomes and lysosomes (*Marquer et al., 2016*). During *C. trachomatis* infection, disruption of CI-MPR association with SNX5 may perturb retromer-dependent receptor recycling, which could disrupt lysosomal function (*Riederer et al., 1994*). The role of lysosomes in *Chlamydia* infection remains incompletely understood; *Chlamydia* avoids fusion with lysosomes but may utilize them as a source of nutrients (*Al-Younes et al., 1999*; *Heinzen et al., 1996*; *Ouellette et al., 2011*; *Rzomp et al., 2003*). Alternatively, since *Chlamydia* acquires cholesterol from the host and since the *Chlamydia* inclusion maintains an intimate association with late endosomal and lysosomal compartments where cholesterol accumulates (*Carabeo et al., 2003*; *Elwell et al., 2016*), retromer dysfunction could enhance cholesterol transport to the inclusion. Altogether, our structural studies of IncE binding to the SNX5-PX domain reveal a highly conserved hydrophobic groove on SNX5 that is required for the interaction of the CI-MPR with retromer, a process that can be targeted by pathogens to avoid retromer-mediated host restriction of vacuolar pathogens.

## Materials and methods

### Protein expression and purification

Mouse 8xHis-SNX5-PX$_{20-180}$ (UniProt Q9D8U8), which is 98% identical to human, was expressed in BL21(DE3) *E. coli* cells. Briefly, cells were grown at 37°C to OD$_{600}$ 0.5 in Terrific Broth supplemented with 50 µg/ml kanamycin. Expression of 8xHis-SNX5-PX$_{20-180}$ was induced by addition of 1 mM isopropyl ß-D-thiogalactoside (IPTG) and cells were incubated overnight at 18°C. Cells were pelleted, resuspended in lysis buffer (2.5x PBS (3.7 mM KH$_2$PO$_4$, 38 mM Na$_2$HPO$_4$, 300 mM NaCl, 7 mM KCl), 10 mM imidazole, 5 units/ml Benzonase, and 1 mg/ml lysozyme). Cells were lysed using a microfluidizer (Microfluidics) and the lysate was cleared by centrifugation for 25 min at 15,000 x g. The supernatant was loaded onto a Ni-NTA column (Qiagen) equilibrated in 50 mM Hepes pH 7.4, 500 mM NaCl, 5 mM ß-mercaptoethanol and 20 mM imidazole (buffer A). After washing with 15 column volumes buffer A, the imidazole concentration was increased to 43 mM for two column volumes, and protein was eluted in 50 mM Hepes pH 7.4, 500 mM NaCl, 5 mM ß-mercaptoethanol and 250 mM imidazole. Fractions containing SNX5 were pooled and dialyzed against 50 mM Hepes pH 7.4, 100 mM NaCl, 1 mM TCEP (Tris(2-carboxyethyl)phosphin) for 16 hr at 4°C in the presence of 3C protease. The protein was passed twice over Ni agarose resin (Gold Biotechnology) to remove the protease and purification tag, concentrated and injected onto a Superdex 75 10/300 column (GE

Healthcare Life Sciences) equilibrated in 20 mM Hepes pH 7.4, 100 mM NaCl and 1 mM TCEP (buffer G). SNX5-containing fractions were pooled and concentrated to 31.2 mg/ml. The IncE (Uni-Prot P0DJI4) peptide encompassing residues 108–132 was synthesized by Peptide 2.0 at 90.44% purity and diluted in 20 mM Hepes pH 7.4, 100 mM NaCl and 1 mM TCEP to 3.94 mM. For complex formation, $IncE_{108-132}$ was mixed with purified $SNX5-PX_{20-180}$ at a 3.4:1 (mol/mol) ratio and injected onto a Superdex 75 10/300 size exclusion column equilibrated in buffer G. The SNX5:IncE complex was concentrated to 12.2 mg/ml.

## Crystallization and structure determination

Crystals of the SNX5-PX:IncE complex were grown at 20°C by sitting drop vapor diffusion against 26% (w/v) PEG1500. The SNX5-PX:IncE complex (12.2 mg/ml) was mixed with an equal volume of reservoir solution. Crystals were flash frozen in liquid nitrogen without additional cryoprotection. Diffraction data was collected at the Lawrence Berkeley National Laboratory Advanced Light Source (ALS) Beamline 8.3.1. Diffraction data was processed using mosflm (*Battye et al., 2011*). The structure was determined by molecular replacement with Phaser (*McCoy et al., 2007*) from the PHENIX (*Adams et al., 2010*) suite of programs using the rat SNX5-PX domain (PDB ID 3HPC, residues 30–169) as a search model. Strong positive density peaks in the $F_o$-$F_c$ density map allowed manual placement of an IncE molecule in Coot (*Emsley et al., 2010*). Density fitting with Coot was alternated with refinement in PHENIX using standard refinement protocols (*Adams et al., 2010*). The refinement of the structure was optimized using the PDB_REDO web server (*Joosten et al., 2014*). Data collection and refinement statistics are shown in *Supplementary file 3*. All structure figures were prepared using PyMOL (The PyMOL Molecular Graphics System, Version 1.7.4.4 Schrödinger, LLC). Coordinates have been deposited in the RCSB Protein Data Bank with accession number 5TP1.

## Antibodies and plasmids

Antibodies were obtained from the following sources: mouse anti-SNX1 (BD Biosciences, 611482), rabbit anti-SNX5 (Santa Cruz, H-40), goat anti-SNX6 (Santa Cruz, N-19), goat anti-VPS35 (Imgenex, IMG-3575), mouse anti-FLAG (Sigma, F3165), mouse anti-GAPDH (Millipore, MAB374), rabbit anti-CI-MPR (EPR6599) (Abcam, ab124767), rabbit anti-Strep TagII HRP (Novagen, 71591–3), goat anti-rabbit IgG HRP (BioRad), goat anti-mouse IgG HRP (BioRad), anti-goat IgG HRP (BioRad). Secondary fluorescent antibodies were derived from donkey (Life Technologies): anti-goat Alexafluor 647, anti-mouse Alexafluor 568, anti-rabbit Alexafluor 488. Rabbit anti-IncE antibody was kindly provided by Ted Hackstadt (Rocky Mountain Laboratories, NIH).

The plasmid encoding FLAG-tagged mouse SNX5 was a kind gift from Jia-Jia Liu (Chinese Academy of Sciences) (*Hong et al., 2009*). The plasmid encoding human SNX15 was a kind gift from Michael Starnbach (Harvard) and was shuttled to a vector encoding a C-terminal FLAG tag by Gateway cloning. The region encoding $SNX5-PX_{20-180}$ was PCR amplified and inserted into the SmaI site of pH3C-LIC using ligation independent cloning to generate a protein with an N-terminal 8xHis fusion. The SNX5 single (Y132D, Y132N, F136D, F136N) and double point mutants (Y132D, F136D and Y132N, F136N) were generated by site directed mutagenesis (QuickChange, Agilent) using the template plasmid encoding FLAG-SNX5. Construction of the plasmid encoding $IncE_{101-132}$-Strep or 6xHis-MBP-$IncE_{101-132}$-Strep was previously described (*Mirrashidi et al., 2015*). Primers are listed in *Supplementary file 4*.

## Cell culture and bacterial propagation

HeLa cells (obtained from American Type Culture Collection) and HEK293T (a generous gift from NJ Krogan) cells were maintained at 37°C with 5% $CO_2$ in Eagle's Minimum Essential Media (MEM) or Dulbecco's modified Eagle's medium (DMEM) containing 10% fetal bovine serum (FBS) respectively. Cells were tested as mycoplasma free. *C. trachomatis* serovar D (UW-3/Cx) was propagated as previously described (*Elwell et al., 2011*). Plasmid transfections were performed using Effectene (Qiagen) or Continuum (Gemini Bio Products) for 24 or 48 hr following manufacturer's instructions.

## Lipid binding assay

Protein:lipid binding assays were performed using commercially available PIP strips (Molecular Probes). Briefly, membranes spotted with lipids were blocked with TBST (50 mM Tris-HCl pH 7.4 and

150 mM NaCl, 0.1% Tween) containing 3% fatty-acid free bovine serum albumin (Sigma) for 1 hr at room temperature. Membranes were then incubated with purified SNX5-PX$_{20-180}$ or SNX5-PX$_{20-180}$ in complex with IncE$_{108-132}$ peptide for 1.5 hr and then washed with TBST. Bound SNX5 was detected by immunoblot analysis with a rabbit anti-SNX5 antibody.

## Affinity purification and mass spectrometry

Affinity purifications were performed as previously described (*Jäger et al., 2011b*; *Mirrashidi et al., 2015*). Briefly, HEK293T cells were seeded in 10 cm$^2$ plates, and transfected the next day with 6 µg/ plate of the indicated purified plasmid DNA using Effectene transfection reagent (Qiagen). At 48 hr after transfection with the full-length FLAG-tagged SNX5$_{WT}$, SNX5$_{Y132D}$, or SNX5$_{F136D}$ encoding plasmids, cells were detached with 10 mM EDTA/D-PBS, washed with PBS and lysed with 1 ml of ice cold Final Wash buffer (50 mM Tris-HCl pH 7.5, 150 mM NaCl, 1 mM EDTA) plus 0.5% NP-40, Roche Complete protease and PhosSTOP phosphatase inhibitor. Lysates were incubated with 30 µl of FLAG beads (Sigma) and incubated overnight, rotating at 4°C. Beads were washed three times in Final Wash Buffer plus 0.05% NP-40, and then once in Final Wash Buffer. Samples were eluted in 45 µl of 100 µg/ml FLAG peptide (Sigma) in Final Wash Buffer. All AP-MS were performed in triplicate and assayed by Immunoblot using chemiluminescence (Denville Scientific) or by silver stain (Pierce). Purified protein eluates were digested with trypsin for LC-MS/MS analysis and processed as previously described (*Mirrashidi et al., 2015*). Digested peptides were analyzed on a Thermo Fisher Orbitrap Fusion mass spectrometry system equipped with a Easy nLC 1200 ultra-high pressure liquid chromatography system interfaced via a Nanospray Flex nanoelectrospray source. Samples were injected on a C18 reverse phase column (25 cm x 75 um packed with ReprosilPur C18 AQ 1.9 um particles). Peptides were separated by an organic gradient from 5% to 30% ACN in 0.1% formic acid over 60 minutes at a flow rate of 300 nl/min. The MS continuously acquired spectra in a data-dependent manner throughout the gradient, acquiring a full scan in the Orbitrap (at 120,000 resolution with an AGC target of 200,000 and a maximum injection time of 100 ms) followed by as many MS/ MS scans as could be acquired on the most abundant ions in 3s in the dual linear ion trap (rapid scan type with an intensity threshold of 5000, HCD collision energy of 29%, AGC target of 10,000, a maximum injection time of 35 ms, and an isolation width of 1.6 m/z). Singly and unassigned charge states were rejected. Dynamic exclusion was enabled with a repeat count of 1, an exclusion duration of 20 s, and an exclusion mass width of +/- 10 ppm.

Raw mass spectrometry data were assigned to human protein sequences and MS1 intensities extracted with the MaxQuant software package (version 1.5.5.1) (*Cox and Mann, 2008*). Data were searched against the SwissProt human protein database (downloaded on January 11, 2016). Variable modifications were allowed for N-terminal protein acetylation, methionine oxidation. A static modification was indicated for carbamidomethyl cysteine. The 'Match between runs' feature was enabled to match within 2 minutes between runs. All other settings were left using MaxQuant default settings. The intensity information from the MS system was searched using freely available MaxQuant proteomics software. The spectral counts for the identified peptides were then analyzed using SAIN-Texpress (http://saint-apms.sourceforge.net/Main.html) to determine the confident protein-protein interactions in the set based on theBFDR. Interactions with a BFDR $\leq$ 0.1 were considered significant and included in *Table 1* in the main text.

## IncE peptide competition assay

HEK293T cells were transfected with FLAG-SNX5 for 48 hr, lysed in Final Wash Buffer containing 0.5% NP-40 and immunoprecipitated with anti-FLAG magnetic beads (Sigma) in the presence of the indicated concentrations of IncE wild type or mutant peptide. Beads were washed and proteins were eluted with FLAG peptide as described above. Eluates were subjected to Immunoblot analysis with the indicated antibodies. IncE wild type and mutant peptides corresponding to amino acids 108–132 were synthesized by Peptide 2.0 at crude purity and resuspended in Final Wash Buffer to 4.02 mM. IncE wild type peptide sequence is PANEPTVQFFKGKNGSADKVILVTQ and IncE$_{V114N,F116D}$ mutant peptide sequence is PANEPTNQDFKGKNGSADKVILVTQ.

## In vitro pull-downs

Expression and purification of 6xHis-MBP-IncE$_{101-132}$-Strep was performed as previously described (*Mirrashidi et al., 2015*). Purified 6xHis-MBP-IncE$_{101-132}$-Strep was immobilized on Strep-Tactin Sepharose beads (IBA) and incubated with purified 8xHis-SNX5-PX$_{20-180}$ for 1 hr, rotating at 4°C in binding buffer (50 mM Tris-HCl pH 7.5, 150 mM NaCl), in the presence of the indicated concentrations of IncE wild type or mutant peptide. Beads were washed and proteins were eluted with 2.5 mM D-desthiobiotin (Invitrogen). Eluates were subjected to immunoblot analysis with the indicated antibodies.

## Co-immunoprecipitation

HEK293T cells were co-transfected with a plasmid encoding IncE$_{101-132}$-Strep and either full-length FLAG-tagged SNX15$_{WT}$, SNX5$_{WT}$, or SNX5 mutants for 48 hr. Cells were lysed in Final Wash Buffer containing 0.5% NP-40, and proteins were immunoprecipitated with FLAG beads followed by elution with FLAG peptide. Eluates were subjected to Immunoblot analysis with the indicated antibodies. For co-immunoprecipitation during infection, HeLa cells were transfected with full-length FLAG-SNX5$_{WT}$ for 24 hr and infected with *C. trachomatis* serovar D for 1 hr. At 24 hr post-infection (hpi), cells were lysed in Final Wash Buffer containing 0.5% NP-40 and proteins were immunoprecipitated with FLAG beads followed by elution with FLAG peptide. Eluates were subjected to Immunoblot analysis with the indicated antibodies. Band intensity was quantified using ImageJ (*Schneider et al., 2012*).

## Microscopy

HeLa cells were grown on glass coverslips, transfected with full-length FLAG-tagged SNX5$_{WT}$ or SNX5$_{Y132D,F136D}$ and infected with *C. trachomatis* for 24 hr. Cells were fixed in 4% paraformalde-hyde/D-PBS, permeabilized in 1x D-PBS containing 0.1% Triton X-100 for 5 min at room temperature, blocked in 1x/D-PBS containing 1% BSA for 1 hr, and stained with the indicated primary and fluorophore-conjugated secondary antibodies in blocking buffer for 1 hr each. Coverslips were mounted in Vectashield mounting media containing DAPI (Vector Laboratories) to identify bacteria and host cell nuclei. Single z slices were acquired using Yokogawa CSU-X1 spinning disk confocal mounted on a Nikon Eclipse Ti inverted microscope equipped with an Andora Clara digital camera and CFI APO TIRF 60x oil objective. Images were acquired by NIS-Elements software (Nikon). The exposure time for each filter set for all images was identical. Images were processed with Adobe Photoshop CS and NIS-Elements.

## Bioinformatics

A BLAST search (*Altschul et al., 1990*) was performed using *M. musculus* SNX5 as input. The resulting sequences were culled to remove all partial sequences. Sequences that were titled 'Sorting nexin-like', leaving 1273 sequences, were then aligned in CLUSTAL Omega (*Sievers et al., 2011*) and displayed in ESPript (http://espript.ibcp.fr). Sequences displayed in the figure were chosen manually to display the diversity of SNX sequences but are extracted from the larger alignment. To determine highly conserved residues in the SNX5/6/32 family, we further culled the list to include only sequences with the terms: 'sorting nexin-5' or 'sorting nexin-6' or 'sorting nexin-32'. These were realigned in CLUSTAL Omega, partial sequences that did not align in the PX domain were removed manually and the remaining 763 sequences were realigned. Only 13 residues were found to be completely invariant across all sequences, as annotated in *Figure 2*.

## Acknowledgements

We thank Drs. Raphael Validivia, Jia-Jia Liu, Ted Hackstadt, and Michael Starnbach for reagents and members of the Engel, Rosenberg, and Krogan lab for support and advice. We thank Adam Frost, Mark von Zastrow, James Holton and Rushika Perera for helpful discussions. The authors gratefully acknowledge financial support from the National Institutes of Health, the University of California, San Francisco Gladstone Institutes for Virology and Immunology, and the Center for AIDS Research. ALS is supported by the US. DOE Contract DE-AC02-05CH11231 and we appreciate the support of Jane Tanamachi and George Meigs.

## Additional information

### Funding

| Funder | Grant reference number | Author |
|---|---|---|
| National Institutes of Health | K08AI091656 | Oren S Rosenberg |
| National Institutes of Health | AI073770 | Joanne N Engel |
| National Institutes of Health | AI105561 | Joanne N Engel |

The funders had no role in study design, data collection and interpretation, or the decision to submit the work for publication.

### Author contributions

CAE, NC, Conceptualization, Formal analysis, Investigation, Methodology, Writing—original draft, Writing—review and editing; JvD, Data curation, Formal analysis, Methodology; JRJ, RN, KM, Investigation, Methodology; NJK, Conceptualization, Resources, Data curation, Supervision, Funding acquisition, Project administration; JNE, Conceptualization, Formal analysis, Supervision, Funding acquisition, Writing—original draft, Project administration, Writing—review and editing; OSR, Conceptualization, Formal analysis, Supervision, Investigation, Methodology, Writing—original draft, Writing—review and editing

### Author ORCIDs

Cherilyn A Elwell, http://orcid.org/0000-0001-7702-3938
Nadine Czudnochowski, http://orcid.org/0000-0002-0771-0721
Oren S Rosenberg, http://orcid.org/0000-0002-5736-4388

## Additional files

### Supplementary files

• Supplementary file 1. MaxQuant output for affinity purification experiments described in *Figure 3*. Files are organized by the 'RawFile' names and describe LC-MS/MS runs recorded from three biological replicates. The file names are as follows: mock interactions (RawFiles FU20160603-24, FU20160603-32 and FU20160603-02), WT SNX5 interactions (RawFiles FU20160603-22, FU20160603-26m, and FU20160603-04) $SNX5_{Y132D}$ interactions (RawFiles FU20160603-20, FU20160603-34, and FU20160603-06) and $SNX5_{F136D}$ (RawFiles FU20160603-18, FU20160603-28, and FU20160603-08).

• Supplementary file 2. Summary of results from SNX5 affinity purification and mass spectrometry. Columns C-N show the number of peptide-charge state pairs (Num Pep/Charge) and columns O-Z show the sum of all peptide intensities (Intensity) for each protein detected in each biological replicate as indicated. Proteins are identified by UniProt accession numbers; entries with multiple UniProt accession numbers represent those identified by peptide sequences shared by multiple protein sequences.

• Supplementary file 3. Data collection and refinement statistics.

• Supplementary file 4. List of primers.

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
