## [Decision Letter]

Thank you for submitting your article "Chlamydia interfere with binding of the mannose-6-phosphate receptor to sorting nexins to counteract host restriction" for consideration by *eLife*. Your article has been favorably evaluated by Vivek Malhotra (Senior Editor) and three reviewers, one of whom, Suzanne R Pfeffer, is a member of our Board of Reviewing Editors. The following individuals involved in review of your submission have agreed to reveal their identity: Oliver Daumke (Reviewer #2) and Volker Haucke (Reviewer #3).

The reviewers have discussed the reviews with one another and the Reviewing Editor has drafted this decision to help you prepare a revised submission.

The transmembrane protein IncE from Chlamydia is incorporated into a membrane-bound compartment containing the bacteria, the inclusion, with the N- and C-terminal moieties pointing towards the cytosol. In proteomic screens, the authors of this study have previously found that the C-terminal portion of IncE interacts with the PX domain of SNX5, leading to missorting of retromer. In the current study, they report the crystal structure of the IncE-SNX5 complex. IncE forms an antiparallel hairpin structure that binds as an extension of the SNX5-PX domain β-sheet. The authors confirm their structure by mutagenesis and pull-down experiments. They find that the IncE interaction site in the SNX5-PX domain is evolutionary highly conserved, suggesting a conserved cellular function. Using mass spectrometry, they identify the known retromer cargo cation-independent mannose-6-phosphate receptor (CI-MPR) as binding partner of the SNX5 PX domain. They then show that addition of an IncE peptide, but not of an IncE variant peptide with mutations in the PX domain binding interaction site, disturbs the SNX5-CI-MPR interaction in cell lysates. In the final experiment, they demonstrate that Chlamydia infection interferes with binding of CI-MPR to SNX5.

The manuscript is well written and easy to follow. It describes the first structural characterization of a Chlamydia Inc protein with its cellular target, therefore elucidating a molecular mechanism how Chlamydia survives in its host environment; it is therefore of general interest to researchers of several disciplines. Most of the experiments are convincing. However, the story would be significantly enhanced by extending the structural and mass spectrometry analyses to better characterize the interaction of Cl^-^MPR with IncE. Given that the CI-MPR cytoplasmic domain is only 163 residues, it should not require much work to synthesize peptides covering that region to determine which residues bind to SNX5, to complete this satisfying and high quality story.

Structural analyses:

In the validation report, four SNX5-IncE complexes are mentioned. Are they identical or do they show structural variations? What exactly are the structural changes of the SNX5 domain between the liganded and unliganded state? Please provide some more figures.

Figure 1 – What was the algorithm to generate this surface? What do the green and yellow colors exactly mean? Are they really representing hydrophobicity?

Mass spectrometry:

The mass spectrometry data need to be further analyzed and better represented, rather than providing protein lists in excel sheets. Is Cl^-^MPR the only identified interaction partner or which classes of proteins have been identified as specific interactors? Are there other known retromer cargoes or components in the list? Can some of them be independently confirmed? Again, some additional figures or clarification will help.

Interaction of CI-MPR with SNX5.

It is important to demonstrate that the IncE interface in SNX5 is the direct binding site of cargo, such as CI-MPR, which was also reported to bind to the Vps35 subunit (Arighi, JCB 2004). The authors should finish and include their ongoing experiments to find out what is the direct interaction partner of SNX5.

Other comments:

1) Some of the key biochemical data are qualitative (dot blots and Figure 2). Please try to be quantitative wherever possible. We suggest that you remove the dot blots or move them to supplemental because they are not relevant to the main part of your story and are only initial screens of PIP binding capacity and specificity. If included as supplemental, please use tentative wording in terms of interpretation.

2) No information is presented on the effect of SNX5 mutations with respect to its localization to endosomes vs. the bacterial inclusion.

3) Based on the model one would expect structural similarity between the CI-MPR tail and IncE. Has this been looked at? Moreover, how do the authors envision competition between IncE and CI-MPR to occur? Is this via mass action due to excessive production of IncE or is the affinity of SNX5 higher for IncE compared to the MPR?

---

## [Author Response]

*[…] The manuscript is well written and easy to follow. It describes the first structural characterization of a Chlamydia Inc protein with its cellular target, therefore elucidating a molecular mechanism how Chlamydia survives in its host environment; it is therefore of general interest to researchers of several disciplines. Most of the experiments are convincing. However, the story would be significantly enhanced by extending the structural and mass spectrometry analyses to better characterize the interaction of Cl^-^MPR with IncE. Given that the CI-MPR cytoplasmic domain is only 163 residues, it should not require much work to synthesize peptides covering that region to determine which residues bind to SNX5, to complete this satisfying and high quality story.*

*Structural analyses:*

*In the validation report, four SNX5-IncE complexes are mentioned. Are they identical or do they show structural variations?*

There are four copies of the SNX5-IncE complex in the asymmetric unit. These are not significantly different, at least not in ways that can’t be explained by the subtle differences in the chemical environment of each protein chain. An overlay of the four copies has been provided as a supplemental figure (Figure 1—figure supplement 1). We have also added to the text a clarification that our discussion of the structure is based on observations of chain A and its associated IncE (chain P), although, as shown in our new figure, there does not appear to be a significant difference between the chains.

The legend for the new figure reads:

*“*Figure 1—figure supplement 1. Structural analysis of the SNX5-PX:IncE complex.

A. Overlay of the four SNX5-PX:IncE complexes present in the asymmetric unit. The B-factors of the Cα atoms are indicated by colors along a spectrum as shown in the figure.”

*What exactly are the structural changes of the SNX5 domain between the liganded and unliganded state? Please provide some more figures.*

The changes with binding of IncE are small, mainly consisting of an inward rotation of the SNX5 α-helical extension ~ 30°toward the IncE peptide. It remains possible that there is a downstream propagation of this change to a functional output. However, we cannot exclude that this rotation is simply a function of crystal contacts in a relatively flexible region of the protein. We now provide an overlay of the SNX5-PX:IncE complex with the apo-SNX5-PX structure in Figure 1—figure supplement 1.

The legend for the new figure reads:

*“*B.Overlay of the SNX5-PX:IncE complex with the apo-SNX5-PX core structure (PDB ID 3HPC). The structures are colored by their root-mean-square deviation (RMSD) indicated by a color gradient from blue to red. The average RMSD is 1.77 Å. Molecules are rotated 75° around the y-axis compared to panel A.”

*Figure 1 – What was the algorithm to generate this surface? What do the green and yellow colors exactly mean? Are they really representing hydrophobicity?*

We have used pymol to remake this figure with a more informative color coding scheme, as described in (doi:10.3389/fmolb.2015.00056). We have supplied a new figure (Figure 1) and explained the color coding in the legend. There are no charged residues at the IncE/SNX5 interface.

The legend for the new Figure 1 reads:

“B.Surface and ribbon representation of SNX5-PX showing the hydrophobic binding groove. […] The IncE interacting residues are shown as sticks and labeled. IncE is displayed as gold ribbon”

*Mass spectrometry:*

*The mass spectrometry data need to be further analyzed and better represented, rather than providing protein lists in excel sheets. Is Cl^-^MPR the only identified interaction partner or which classes of proteins have been identified as specific interactors? Are there other known retromer cargoes or components in the list? Can some of them be independently confirmed? Again, some additional figures or clarification will help.*

In the revised paper, we have provided a new table of results and statistics (Table 1). The methods we used are also further clarified in the text, as follows:

“Using SAINT-Express (Teo, 2014), we compared the protein-protein interactions of affinity purified lysates from untransfected cells to those prepared from SNX5_WT_, SNX5_Y132D_, or SNX5_F136D_ transfected cells. […] We focused on CI-MPR because it is the most well-studied retromer cargo and because we have previously established that ectopic expression of IncE is sufficient to disrupt CI-MPR trafficking (Mirrashidi, 2015).”

The legend for the new Table 1 reads:

“Table 1: SNX5 protein-protein interactions and their dependence on the IncE binding groove.[…] Of the proteins found to interact with SNX5_WT_, CI-MPR and IGF1R show a difference in their SAINT score between the WT and the mutants, suggesting that the mutations block the protein-protein interaction.”

Interaction of CI-MPR with SNX5.

*It is important to demonstrate that the IncE interface in SNX5 is the direct binding site of cargo, such as CI-MPR, which was also reported to bind to the Vps35 subunit (Arighi, JCB 2004). The authors should finish and include their ongoing experiments to find out what is the direct interaction partner of SNX5.*

We provide strong evidence that CI-MPR and SNX5 interact and that this interaction is disrupted by the binding of SNX5 to IncE, either *in vitro* or with infection. These data are consistent with our earlier finding that overexpression of IncE and subsequent binding of SNX5/6 disrupts normal trafficking of CI-MPR. As the reviewers rightly indicated, our findings do not prove the interaction between CI-MPR and SNX5 is direct. This point is especially important because CI-MPR has been demonstrated to interact with VPS35, another component of the retromer. A direct interaction of CI-MPR and VPS35 has been demonstrated by yeast two hybrid studies. Since SNX5/6 is in the cytosol, we assume that any interaction between SNX5/6, either direct or indirect, would be mediated by the cytosolically exposed C-terminal region of CI-MPR. We have pursed several methods to purify the C-terminus of CI-MPR for use in direct *in vitro* binding studies. First, we purified CI-MPR fused to maltose binding protein (MBP) but unfortunately this construct yielded protein that was likely insoluble and/or improperly folded, making it impossible to interpret binding studies. We then purified CI-MPR fused to glutathione-S-transferase (GST), but the yields were low and highly contaminated with free GST from cleavage events. In general, we had significant issues with proteolysis of the CI-MPR.

We used three different assays with these purified proteins to try to demonstrate a direct interaction between SNX5 and CI-MPR. First, we attempted pulldown experiments using immobilized GST-CI-MPR. As this was not successful possibly due to high background binding, we were concerned this was a weak interaction that was disrupted by the washing steps in the pull-down protocol. Therefore, we took a subtractive approach that did not rely on washing. This also was not successful, despite extensive trouble- shooting. Next, we further purified away as much free GST as possible and bound the GST-CI-MPR to the sensor tip on an Octet. Again, we observed substantial non-specific binding, even in the presence of BSA and detergent. Troubleshooting the technical issues encountered in our binding assays are beyond the scope of this paper. We note that our experimental approach might not have been successful for the additional following reasons:

1) Binding requires the presence of or is mediated by another protein. This partner is present when we do pull-downs from cells but not detectable by MS;

2) Binding requires a post-translational modification that is not present in bacterially produced purified protein;

3) Binding is direct, but our constructs do not contain the correct resides to allow for binding or the correct residues may have been lost due to proteolysis.

Although we strongly favor the hypothesis that CI-MPR binds directly to the hydrophobic binding domain of SNX5/6, at this time we cannot prove a direct interaction between CI- MPR and SNX5/6. However, we still believe we have made an important and original contribution about the interaction by showing: 1) CI-MPR interacts with SNX5 in a VPS35-independent manner (as shown by western blot and mass spectrometry), 2) Interactions between CI-MPR and SNX5 are dependent on the same hydrophobic groove that is required for direct binding of SNX5/6 to IncE and 3) the interaction between CI-MPR and SNX5 can be competed away by addition of wild type, but not mutant, IncE, both during an affinity purification experiment and in the context of infection, showing that one effect of IncE expression is to delocalize SNX5 from CI-MPR.

*Other comments:*

*1) Some of the key biochemical data are qualitative (dot blots and Figure 2). Please try to be quantitative wherever possible. We suggest that you remove the dot blots or move them to supplemental because they are not relevant to the main part of your story and are only initial screens of PIP binding capacity and specificity. If included as supplemental, please use tentative wording in terms of interpretation.*

While we concur that the PIP strip experiment is qualitative, we believe that this data provides important confirmatory data that IncE interacts with a SNX5 surface distinct from the PI binding site. As in our initial submission, this data is provided as supplemental data (Figure 1—figure supplement 2). We have revised the text as follows:

“In the SNX5-PX_20-180_:IncE_108-132_ crystal structure, we observed no direct interaction between IncE_108-132_ and the previously observed PIP binding pocket (Figure 1—figure supplement 3). […] Together, this data supports the notion that binding of IncE to SNX5-PX does not interfere with SNX5 binding to PIPs.”

In the Discussion, we have revised the text as follows:

“However, our structure revealed no major conformational changes in the PX domain or the BAR domain-interacting surface. This finding is consistent with our observation that IncE peptide does not interfere with SNX5 binding to immobilized phospholipids or to SNX1.”

*2) No information is presented on the effect of SNX5 mutations with respect to its localization to endosomes vs. the bacterial inclusion.*

We have now included a new figure (Figure 2—figure supplement 1) in which we compare the localization of transfected FLAG-SNX5WT and SNX5_Y132D,F136D_ in *Chlamydia*-infected cells. Whereas wild type FLAG-SNX5 is robustly recruited to the inclusion, FLAG-SNX5_Y132D,F136D_ fails to localize to the inclusion.

We have edited the text as follows:

“We previously demonstrated a robust recruitment of full-length FLAG-SNX5_WT_ to the *C. trachomatis* inclusion and associated tubules (Mirrashidi, 2015). We next determined whether the IncE binding groove is necessary for recruitment of SNX5 to the *C. trachomatis* inclusion by analyzing the localization of FLAG-SNX5_Y132D, F136D_ during infection. FLAG-SNX5_Y132D,F136D_ fails to localize to the inclusion (Figure 2—figure supplement 1), although endogenous SNX6 was still recruited. Together, these data provide further support that IncE interacts with the SNX5 hydrophobic binding groove.”

*3) Based on the model one would expect structural similarity between the CI-MPR tail and IncE. Has this been looked at? Moreover, how do the authors envision competition between IncE and CI-MPR to occur? Is this via mass action due to excessive production of IncE or is the affinity of SNX5 higher for IncE compared to the MPR?*

The C-terminal tail of CI-MPR is predicted to be “unstructured”, but contains hydrophobic elements that could potentially mimic IncE. Without a direct binding assay or a structural model, it would be highly speculative to comment on the mechanisms of binding between CI-MPR and SNX5.